# The metabolic underpinnings of sebaceous lipogenesis
Maria Schmidt [1], Hans Binder [1,2] & Marlon R. Schneider [3] ✉

Sebaceous glands synthesize and secrete sebum, a mélange of lipids and other cellular products that safeguards the mammalian integument. Differentiating sebocytes delaminate from the basal membrane and dislodge towards the gland's middle, where they eventually undergo a poorly understood death mode in which the whole cell becomes a secretion product (holocrine secretion). Supported by recent transcriptomics data, this review examines the idea that peripheral sebocytes have a remarkable ability to draw nutrients from the blood and become committed to unrestrainedly invest all available resources into synthetic processes for accomplishing sebum synthesis, thereby exploiting core metabolic fluxes as glycogen turnover, glutamine-directed anaplerosis, the pentose phosphate pathway and de novo lipogenesis. Finally, we propose that metabolic-driven processes are an important mechanistic component of holocrine secretion. A deeper understanding of these metabolic adaptations could indicate novel strategies for modulating sebum synthesis, a key pathogenic factor in acne and other skin diseases.

The white adipose tissue (WAT) is amazingly efficient in storing energy in the form of lipids when food is abundant, and in mobilizing it for daily needs at bad times[1]. Skin sebaceous glands (SG) also focus on the synthesis of lipids, but with the goal of secreting an oily product (sebum) with numerous well-established and putative functions (Fig.1a). While obesity, the excessive fat accumulation potentially impairing health[2], is a serious public health concern and a major focus of current biomedical research, SGs are the research object of a rather few *aficionados* and quite the opposite of a mainstream research area. This disparateness may in part be explained by the sheer differences in the total mass between SG and adipose tissue cells of about three orders of magnitude (a few grams versus about 10 kilograms, respectively) (Fig. 1b). Another reason, as we will address below, is that the functions of the SG and sebum, particularly in humans, remain controversial.

SGs focus essentially on two processes. The first is the constant replacement of mature, disintegrating sebocytes by progenitor cells. The exact molecular mechanisms governing SG cellular turnover remain to be elucidated, and multiple cellular models for SG maintenance have been proposed[3]. Be that as it may, the immediate sebocyte progenitors during homeostasis are proliferating cells that localize to the outermost layer of the gland, where they attach to the basement membrane. Committed sebocytes leave the basal compartment and undergo a morphologically well-defined differentiation program that culminates in holocrine secretion (see below). Adhesion to the basement membrane and metabolism cooperate to control progenitor cell behavior, as recently illustrated by the embigin-regulated localization of the monocarboxylate transporter SLC16A1 in SG basal cells[4]. The second and metabolically determining process is the massive synthesis of various lipid classes for sebum synthesis. For this, committed sebocytes operate a wide portfolio of catabolic and anabolic pathways and seem to invest all available resources into synthetic processes. In particular, their ability to take up and process nutritive substrates from the blood is remarkable. For instance, cytokine-stimulated sebum overproduction was shown to cause weight loss in mice fed with normal or highly calorific diets, to reverse obesity in already overweight animals, and to improve their metabolic parameters[5]. In this review, after a brief introduction to SG physiology, we will focus on sebocyte-typical catabolic and anabolic specializations for acquiring and processing substrates, and examine how exactly these features eventually determine cell downfall.

## Sebaceous glands and sebum

By forming a physical, chemical, and biological protective boundary with the external environment, the mammalian skin fulfills essential functions. It includes the upper epidermis, a multilayered and constantly renewed stratified epithelium, and the lower dermis, in which structures such as hair follicles, sebaceous, and sweat glands are embedded in a fibroblast-rich extracellular matrix also containing nerves and blood vessels. Skin SGs (Fig. 2a) are thus dermal, hair follicle-associated exocrine glands, whose lipid-rich secretion (sebum) lubricates and protects the integument, also by providing skin commensals with an adequate milieu, in addition of having putative antimicrobial and antioxidative functions[6]. Several hormones, as

[1]Interdisciplinary Institute for Bioinformatics (IZBI), University of Leipzig, Leipzig, Germany. [2]Armenian Bioinformatics Institute (ABI), Yerevan, Armenia. [3]Institute of Veterinary Physiology, Veterinary Faculty, University of Leipzig, Leipzig, Germany. ✉e-mail: marlon.schneider@vetmed.uni-leipzig.de

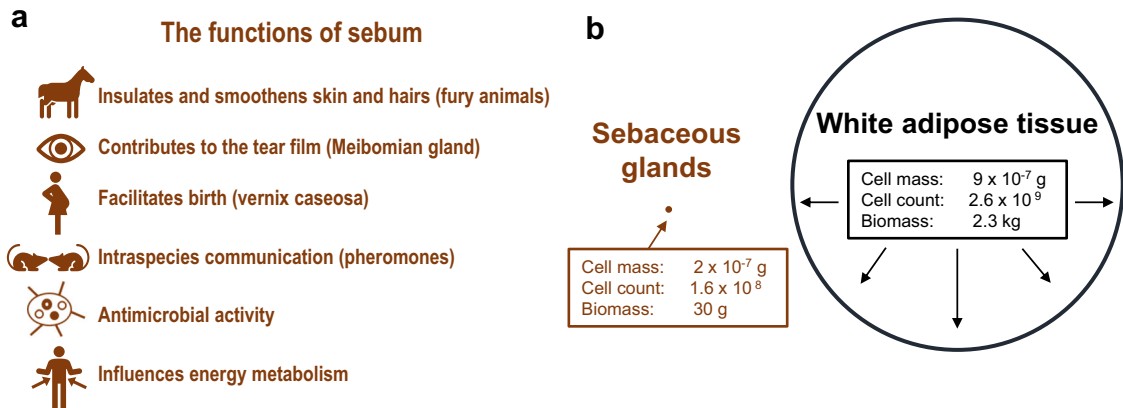

**Fig. 1 | Functions of sebum and the relative dimension of the SG compared to white adipose tissue. a** Summary of established and developing functions of sebum. **b** Comparison of the individual cell mass, cell number, and total mass of SGs and white adipose tissue in adult humans. Based on data from ref. 144.

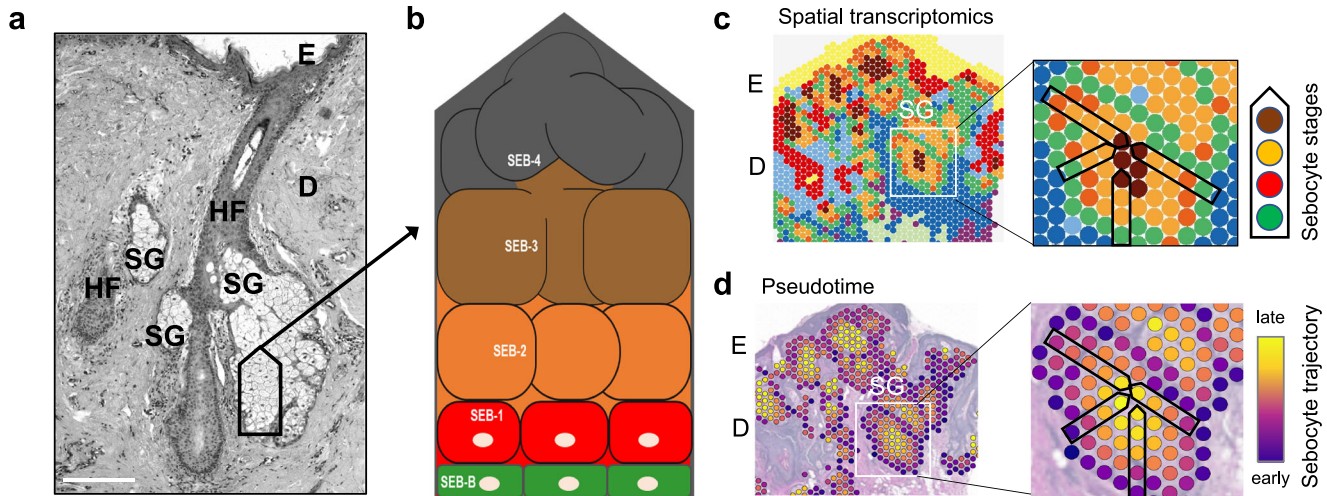

**Fig. 2 | Sebaceous gland histological structure and cellular differentiation.**
**a** Microscopic image of a hair follicle-associated SG. An emerging second hair follicle with its SG can be seen on the left. The scale bar represents 200 μm. **b** Schematic representation of the different sebocyte maturation stages starting from the basal membrane-attached SEB-B via SEB-1 to SEB-4, the latter representing disintegrating sebocytes in the middle of the gland. **c** Spatial transcriptomics images of the human skin; the colors of the spots refer to different cell types (left). An SG in the white framed square is shown enlarged, allowing to discern the different sebocyte maturation stages (right, colors matched to **b**). **d** Pseudotime coloring of the image transfers developmental trajectory information into spatial coordinates (left). Note the concentric, centripetal developmental pattern (right). Images in (**c**, **d**) are adapted from ref. 16. E epidermis, D dermis, HF hair follicle, SG sebaceous gland.

well as the neuroendocrine and the immune systems, regulate sebum secretion[7,8]. Clinical interest in the SG comes from its key role in the pathogenesis of acne, a chronic disease especially prevalent during adolescence that affects billions of people worldwide[9]. Many other debilitating diseases, including some forms of alopecia, atopic dermatitis, or psoriasis display changes in SG structure and/or functions, but their exact pathogenic role remains to be determined[10]. In addition to the hair follicle-associated SGs, specialized "free" SGs are found at specific body sites. Meibomian glands, for instance, are located in the eyelids and produce the sebum-like meibum, an essential part of the tear film[11].

SG acinar cells, called sebocytes, show a progressive differentiation, which begins at the gland's periphery and ends in its middle, when lipid-filled sebocytes disrupt and release their cellular contents, in a process called holocrine secretion[12,13]. Although it is a continuous process, holocrine secretion can be arbitrarily divided in distinct stages according to the cellular maturation degree[14]. Here, we distinguish the following sebocyte (SEB) stages (Fig. 2b): the flat, in part mitotically active cells in contact with the basal lamina at the gland's periphery (SEB-B); the initial (SEB-1) and more advanced (SEB-2) stages of active engagement in lipid synthesis, which includes loss of contact to the basal lamina, massive increase in cell volume

due to cytoplasmic lipid droplet accumulation, and diverse organellar adaptations; and the final maturation stage (SEB-3), which embraces cells poised to or already undergoing disruption. A fifth stage (SEB-4) represents fully lysed and degenerated cells, whose transcriptome cannot be assessed anymore[15,16]. This is a convenient classification as it matches to recent spatial transcriptomics studies to which we will refer again later. The SEB-4 oily debris reaches the skin surface via the sebaceous duct and the hair follicle canal. Because some lipid species are modified by microorganisms and other substances may be picked up underway, sebum composition at the skin's surface differs from that measured in situ immediately after SEB-4 downfall.

Sebum has a peculiar lipid composition, containing triacylglycerides (TAGs), diglycerides and free fatty acids (FFAs) (57%), wax esters (26%), squalene (12%), and cholesterol (2%) in humans[17]. Wax esters and squalene are typical for sebum and are normally not found elsewhere in the body at such high concentrations. Furthermore, sebum FAs show unusual saturation and branching patterns[18]. Sebum lipids show a great diversity across different mammals[19], arguably reflecting functional differences. For instance, equine sebum includes large amounts of giant-ring lactones[20], while human sebum is characterized by the presence of sapienic acid, a C16 FA found nowhere else in the whole animal kingdom[18]. In fury animals,

sebum composition may reflect the specific needs, such as the anti-icing properties of polar bear hairs[21]. In our species, the sebum's functions remain controversial, and a secretion of the embryonic SGs (vernix caseosa) that supports parturition by lubrification of the birth canal and may have additional protective roles, mediates one of the few, if not the only widely acknowledged function of human SGs[22]. Of note, sebum composition differences may be related to the species-specific expression of isoenzymes involved in lipid synthesis. Thus, mouse SGs express a specific glycerol kinase (GK5) that regulates cholesterol biosynthesis independently of cholesterol regulation elsewhere in the body[23], as well as SCD3, a stearoyl-coenzyme A desaturase whose expression is restrict to differentiated sebocytes[24].

## Methodical advances: deciphering SG microphysiology with single cell resolution

As SGs are composed of at most a few hundred sebocytes at various stages of maturation, studying their spatiotemporal physiology requires methods capable of single-cell resolution. The advent of single-cell transcriptomic sequencing, named "Method of the Year" in 2013[25] and, later, spatially resolved transcriptomics, awarded "Method of the Year" in 2020[26], has revolutionized our ability to characterize the human body at single-cell resolution and facilitated the discovery of previously unrecognized cell types and states[27]. The convergence of these cutting-edge molecular and spatial profiling techniques, coupled with novel computational approaches, has provided new insights into SG biology[15,16,28–32]. While these methods do not directly measure metabolites, they enable the reconstruction of underlying gene-regulatory programs with whole-transcriptome resolution in space and time.

Previously, Takahashi and colleagues characterized the anatomical, transcriptional, functional, and pathological profiles of distinct murine epidermal, hair follicle, and hair follicle-associated cell subpopulations, including SGs, offering valuable insights into epidermal and hair follicle differentiation and pathogenesis[32]. Another study, based on the data of Joost and colleagues[33] as well as Cheng and colleagues[34] highlighted intrinsic differences in SG transcripts between mice and humans, underscoring the critical role of peroxisomal processes in this context[29]. Harris and colleagues employed laser microdissection and single-cell RNA sequencing for profiling mouse and human SG transcriptomes[30]. Veniaminova and colleagues explored the effects of skin injury and subsequent healing on SGs, demonstrating that stem cell plasticity contributes to SG resilience following injury[15].

Using spatial transcriptomics, Seiringer and colleagues assessed the SG transcriptome in psoriasis and atopic dermatitis patients, comparing lesional and non-lesional skin samples[28]. The findings revealed that SGs play an active role in skin homeostasis through cell-type-specific lipid metabolism and are not mere bystanders in inflammatory skin diseases; instead, they differentially modulate inflammation in a disease-specific manner. Another study integrated spatial and scRNA-seq datasets and identified two SG clusters, which were not further characterized[35]. In 2024, Schmidt and colleagues presented the first high-resolution spatial portrait of the SG transcriptional landscape[16]. By integrating spatial transcriptomics, pseudotime analysis, RNA velocity, and functional enrichment approaches, the study mapped differentiation in sebaceous hyperplasia and identified novel candidate molecules for regulating SG homeostasis in health and disease. In particular, spatial transcriptomics (Fig. 2c) facilitated the visualization of the SG cell states by mapping transcriptional programs using the average expression levels of genes included in predefined signatures, while bioinformatic analysis organized the transcriptomic spots in a pseudotemporal order based on the similarity of their transcriptional profiles, thereby representing the maturation process along its spatio-temporal dimensions using a continuous pseudotime scale (Fig. 2d). This study, complemented by a publicly accessible online tool, enables in-depth exploration of the spatially resolved SG transcriptome[16].

Collectively, as further discussed below, these studies offer valuable and largely novel insights into the genomic regulation of metabolic pathways that govern sebocyte maturation and sebum production.

## Reporting summary

Further information on research design is available in the Nature Portfolio Reporting Summary linked to this article.

## The metabolic features defining sebaceous glands

Quiescent cells, not needing to conduct overt anabolic functions, rely on a basal rate of glycolysis, converting glucose (and occasionally other substrates as amino acids and FAs) to pyruvate, which is then oxidized in the tricarboxylic acid (TCA) cycle. Proliferating cells, in contrast, have fundamentally different metabolic activities as they require nutrients and energy to duplicate their macromolecular components during each passage through the cell cycle[36,37]. Regarding the carbon flux, the key difference is a much higher glycolysis rate to rapidly generate ATP in the cytoplasm. Part of the generated pyruvate is then converted to lactate (this regenerates $NAD^+$ from NADH, allowing glycolysis to persist), which may be secreted, and part enters the TCA cycle as acetyl-CoA, where several intermediates can be used for macromolecular biosynthesis. Citrate, for instance, is essential for the synthesis of FAs and cholesterol to build up the membrane of daughter cells. Aerobic glycolysis, the preferential fermentation of glucose-derived pyruvate to lactate despite available oxygen, is a key feature of sebocytes engaged in sebum synthesis and will be discussed in detail below. Proliferating cells have, of course, additional metabolic adaptations, including autonomic substrate uptake and processing, and a more versatile usage of substrates as glutamine. Cancer cells frequently harness these pathways to ensure their growth, for instance, by exploiting tumorigenic mutations leading to increased nutrient acquisition and processing[38,39].

Although not mitotically active anymore, sebocytes engaged in lipogenesis definitely have an increased need of energy and building blocks for anabolic processes. They will therefore compulsorily operate a metabolic program that is distinct from that of quiescent or proliferating cells. In the following sections, we will examine the portfolio of sebocyte metabolic adaptations (summarized in Fig. 3) and illustrate our conclusions with data derived from a spatial transcriptomics study (Fig. 4a). A summary of these metabolic processes and their localization along the sebocyte differentiation axis is provided in Fig. 4b.

### Efficient nutrient uptake

While cells can synthesize some of their substrates de novo, they usually rely on extracellular supply of sugars, FAs, amino acids, nucleotides, vitamins, and ions to cover their metabolic requirements, and are thus equipped with multiple transmembrane transporters that allow them to import such nutrients. The transport of hydrophilic solutes as sugars (glucose, but also of other monosaccharides as galactose and fructose), amino acids, and lactate, is mediated by members of the solute carrier (SLC) family. SLCs are one of the largest families of membrane transporters with more than 400 members, and mediate transport either by facilitative (passive passage toward the thermodynamically favorable direction) or secondary active (exploiting the free energy of a coupled, second substrate transported into an energetically favorable direction) mechanisms[40–42]. Notably, different nutrients can utilize the same transporter, and conversely, different SLCs can be employed to transport the same substrate. For instance, members of the SLC16/MCT family are essential for the transport of short-chain monocarboxylates (as lactate, pyruvate, and ketone bodies), hormones, amino acids[43], and SLC16A14 alone has been linked to the uptake of eighteen proteinogenic amino acids[44]. On the other side, glutamine, as the most vital nutrient for proliferating cells after glucose, is transported by several SLCs, whose expression is frequently increased in cancer cells[45]. Also, three SLC classes drive glucose transport into cells: SLC2 members (GLUT family, facilitative transporters), SLC5 members (SGLT family, secondary active transport), and SLC50, which in humans comprise only SLC50A1 (SWEET, a recently described uniporter). This promiscuity confers a remarkable flexibility in obtaining nutrients. Enhanced glucose metabolism is a hallmark of tumors cells[46], and increased glucose uptake is the basis for the wide clinical use of positron emission tomography to detect tumors via their enhanced uptake of a glucose analog[47].

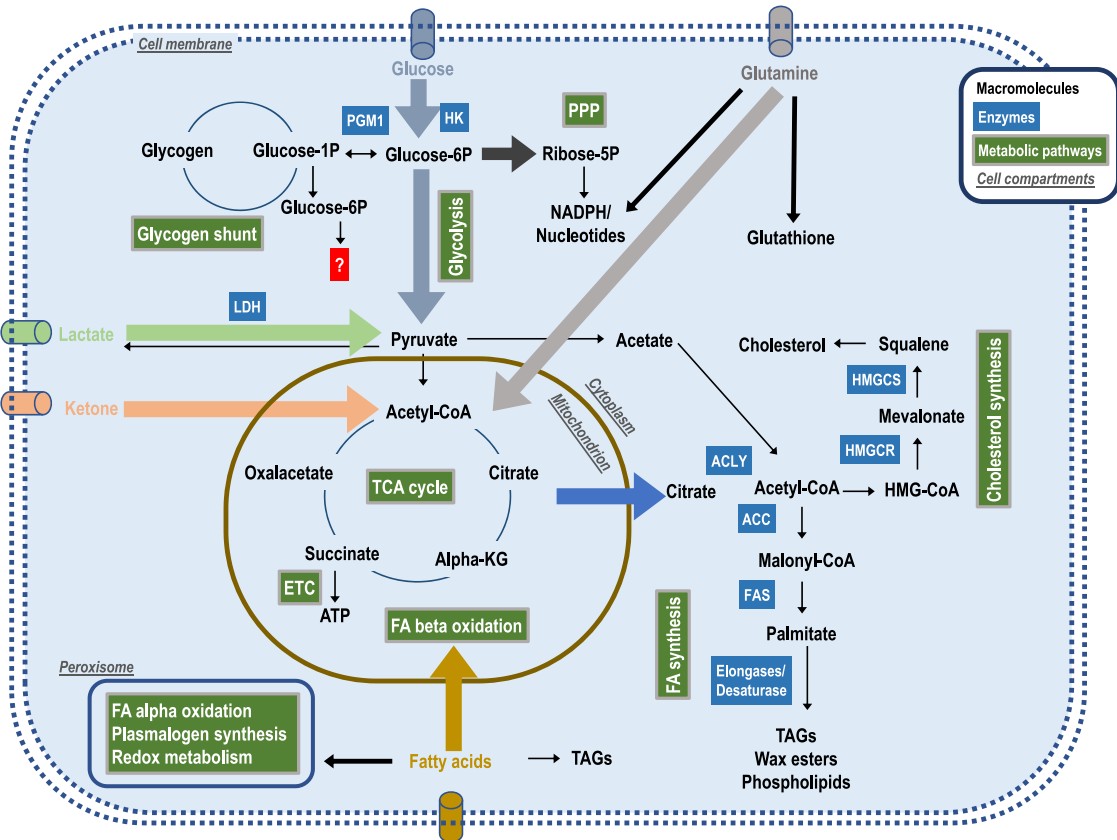

**Fig. 3 | Metabolic pathways governing sebaceous lipogenesis.** Thick arrows represent processes and pathways that may be of particular importance for sebocyte metabolism.

FAs, in contrast, may passively cross cell membranes due to their lipophilic nature, but it is largely accepted that their uptake relies on transport mechanisms. Circulating TAGs are initially hydrolyzed into non-esterified FAs by endothelial lipoprotein lipase (LPL), and their uptake is mediated by a variety of transporters that include the SLC27 family, FA binding proteins, CD36, and caveolins[48]. The short-chain FA acetate may enter cells by the secondary active transporters SLC16A1, SLC16A7, and SLC5A8, as well as by facilitated diffusion via aquaporins[49]. Interestingly, acetate has been identified as a major substrate for central carbon metabolism, especially in nutrient-limited conditions, as pyruvate-derived acetate allows protein acetylation and lipogenesis independently of citrate conversion to acetyl-CoA by ATP-citrate synthase (ACLY)[50]. Upon endothelial TGA hydrolysis, the released FAs enter the cells and can be re-esterified to TAGs or contribute to other lipid species as phospholipids or sphingolipids. Particularly under intense TG hydrolysis, glycerol may become an important substrate. Glycerol diffusion across the cell membrane down the concentration gradient is mediated in humans by four aquaglyceroporins, a subgroup of the aquaporin family of membrane proteins: AQP3, −7, −9, and −10[51]. The enzyme glycerol kinase (GK) converts it into glycerol-3-phosphate, which can serve as a backbone for TG synthesis or enter the gluconeogenic pathway[52]. As intermediates of gluconeogenesis can also be sequestered for anabolic pathways, the importance of glycerol as an energy source remains unknown.

Due to their strategically advantageous location at the SG periphery, SEB-B cells have proper access to a plethora of blood-derived nutrients. Data derived from single cell RNA sequencing indicated abundant expression of SLC family members and other transporters including aquaporins in mouse[33] SG cells, and spatial transcriptomics[16] revealed that human SEB-B cells have strong expression of *SLC1A5*, encoding a key amino acid transporter[45], and of scavenger receptor class B type I (*SCARB1*), encoding a multi-ligand membrane protein receptor that binds to high-density

lipoprotein (HDL) and mediates HDL-dependent cholesterol efflux[53]. Accordingly, the gene set "FA transport" is highly expressed in peripheral SEB-B and SEB-1 stages (Fig. 4a). Also, SLC2A1 (GLUT1), an insulin-independent glucose transporter, is robustly expressed by peripheral sebocytes[54]. Furthermore, altered activity of transporters as SLC1A3[55], SLC27A4/FATP4[56], or FABP5[57] evokes phenotypical changes in SG structure and sebum synthesis (reviewed in ref. 58). Notably, loss of CD36 does not affect the skin, indicating that other transport mechanisms can compensate for CD36 deficiency in the skin[59].

In summary, peripheral sebocytes (and to a lesser degree more centrally located sebocytes) have access to blood-derived nutrients and are equipped with a powerful armamentarium to efficiently take up these substrates to direct them to a variety of anabolic and catabolic pathways.

## Insulin responsiveness and glucagon indifference

The pancreatic hormones insulin and glucagon are arguably the most potent metabolic regulators. Insulin-mediated activation of the insulin receptor induces multiple signaling pathways, including the PI3K/AKT/mTOR and RAS/RAF/MEK pathways, and results in increased uptake of glucose, amino acids, and FAs into cells and promotes synthesis of lipids, proteins, and carbohydrates[60]. The documented and potential effects of insulin on SGs have been extensively examined previously, and include, as we will see below, SG-typical processes as ample nutrient uptake, glycogen synthesis, and de novo lipogenesis[61]. This fits well to the expression of insulin receptor in mouse[33] and human[16] sebocytes. Glucagon, in contrast, has essentially catabolic actions[62], and it inhibits lipid synthesis, the key anabolic activity of SGs. Accordingly, glucagon receptor expression is low or undetectable in sebocytes[16,33].

In summary, sebocyte metabolic activities seem to be greatly influenced by insulin, while potentially counterproductive effects of glucagon signaling are avoided by the absence of corresponding receptors.

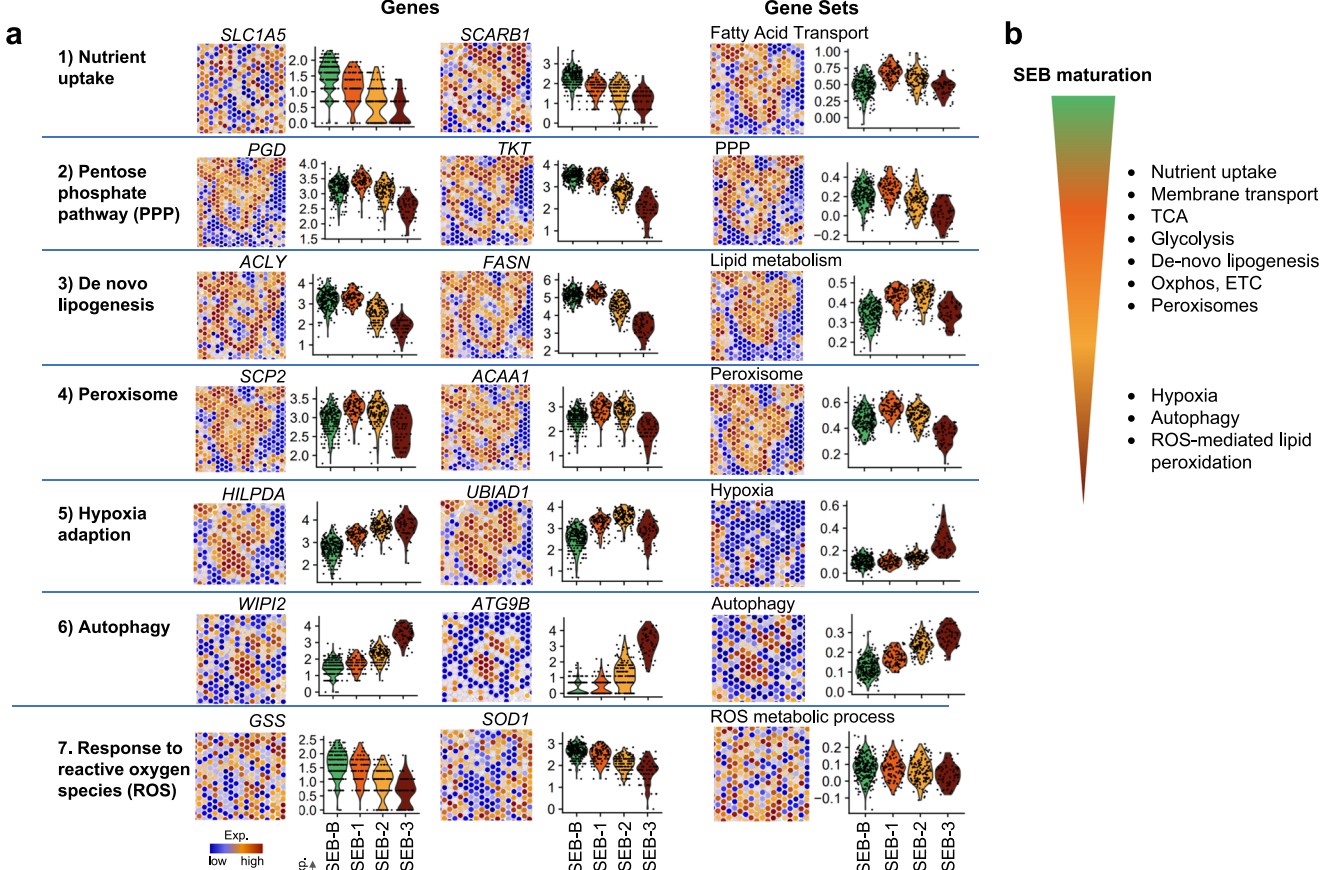

**Fig. 4 | The transcriptional program underlying sebocyte differentiation and sebum synthesis. a** Spatial transcriptomics images (left) and violin plots (right) of individual genes and of gene sets illustrate how the gene expression related to the indicated metabolic processes changes along the human sebocyte differentiation pathway. The "violins" visualize the density distributions of expression values in vertical direction with largest cross-section for maximum point density. Based on data from ref. 16. **b** Overview of the determining metabolic processes governing sebaceous lipogenesis and their position along the sebocyte differentiation pathway.

## Glycogen shunt

Once in the cytoplasm, glucose is phosphorylated to glucose-6-phosphate, which prevents its efflux and facilitates its further usage (Fig. 3). Glycolysis catabolizes glucose to pyruvate, which either enters the mitochondria and fuels the TCA cycle and oxidative phosphorylation or generates lactate. Other possible pathways include the synthesis of serine, glycine, or phosphoglycerol, protein glycosylation, and the pentose phosphate pathway, which supplies the cell with ribose-5-phosphate and NADPH, the sugar base for nucleotide synthesis and an essential antioxidative, respectively.

Glucose can also be temporarily stored as glycogen, which occurs in organ-specific variable amounts. Glycogen is a key biological molecule, as it allows storing glucose as a rapidly accessible carbon cache (Fig. 3). Different isoforms of glycogen synthase (GYS1, GYS2) catalyze the elongation of the glycogen chain, while glycogen phosphorylase isoenzymes (PYGB, PYGL, PYGM) liberate glucose 1-phosphate[63]. Although glycogen is present at high levels in the liver and muscle, the machinery for glycogen turnover is widely expressed across mammalian tissues[64,65]. A recent study revealed that while most circulating lactate is derived from glucose, the majority of glycolytic intermediates in the human body are derived from glycogen[66], highlighting the metabolic importance of this storage form.

The presence of glycogen in SGs, in particular in peripheral sebocytes, was recognized several decades ago[67–71]. In light of the gradual loss of glycogen as lipid amounts increase, early authors already suggested that glycogen may represent an intermediary stage in the transformation of dietary carbohydrates to sebaceous lipids. However, the exact role of glycogen in

sebogenesis has never been uncovered. The routing of glucose carbons through a rapid sequence of glycogen synthesis and degradation, described as the glycogen shunt[72], may confer cells specific metabolic advantages. For instance, intracellular glycogen stores are essential for glycolysis induction and immune effector functions in dendritic cells[73]; in these cells, glucose- and glycogen-derived carbons seem to preferentially contribute to distinct metabolic pathways. Also, inhibition of glycogen breakdown limits viability and proliferation of thyroid cancer cells[65].

Human and mouse sebocytes express predominantly PYGB[29]. PYGB activity is less dependent on hormone signaling and responds more strongly to intracellular nutrient availability[74]. PYGB overexpression has been observed in several aggressive tumors, including colorectal carcinoma, non-small lung cancer, metastatic breast cancer, and anaplastic thyroid cancer[65]. Thus, PYGB expression may contribute to the significant and rapid process of newly imported glucose towards sebaceous lipogenesis or, alternatively, to direct glycogen-derived carbons to an unidentified metabolic pathway.

## The pentose phosphate pathway (PPP) supports anabolism and counteracts oxidative stress

Directing glucose-6P to the PPP[75] provides NADPH for anabolism and for counteracting oxidative stress (the oxidative phase) and generates ribose 5-phosphate, a precursor for the synthesis of nucleotides (the non-oxidative phase) (Fig. 3). The PPP is an efficient way to adapt glucose-derived carbon flux according to the cellular metabolic requirements. Thus, cells focusing on lipid synthesis can adapt their PPP to produce predominantly NADPH. Indeed, SGs strongly engage in the PPP, as illustrated by the activity of the

PPP-related genes phosphogluconate dehydrogenase (*PGD*) and transketolase (*TKT*)[76], as well as by the gene set "PPP" (Fig. 4A).

## Cataplerosis: glycolysis-derived citrate fuels de novo lipogenesis

Once transported into the mitochondria, glycolysis-derived pyruvate is transformed into acetyl-CoA, which standardly follows the ATP-producing TCA cycle and electron transport chain (ETC) pathways (Fig. 3). However, the traditional view of the TCA cycle as a pathway to derive maximal ATP production from oxidizable substrates applies primarily to oxidative but non-proliferative tissues, such as the heart. We propose that in lipogenic sebocytes, similarly to proliferating cells, much of the carbon pool entering the TCA cycle is used in synthetic pathways, resulting in a continuous efflux of intermediates. A prime example of this process (cataplerosis) is the synthesis of lipids. De novo lipogenesis (DNL), the flow of carbons from glucose to lipids[77,78], requires a coordinated sequence of enzymatic reactions (Fig. 3). Briefly, mitochondrial, acetyl-CoA-derived citrate is exported to the cytoplasm, converted to OAA, and then back to the lipogenic precursor acetyl-CoA. Acetyl-CoA can now be directed to the synthesis of FAS via malonyl-CoA by the enzyme ACC, or to the cholesterol synthesis via HMG-CoA by the enzyme HMGCR. The importance of the glycolysis-citrate-lipogenesis pathway in providing the synthetic and bioenergetic requirements that are essential for the growth and proliferation of tumor cells has been convincingly highlighted[79,80]. Indeed, the export of citrate for lipid synthesis may significantly reduce the mitochondrial citrate fraction that is oxidized, resulting in the so-called truncated TCA cycle[81].

It is long known that isolated human SGs perform DNL[82–84], and the activity of enzymes for both FAs and cholesterol synthesis is well documented[85,86]. A more recent and well-elaborated study employing deutered water ($^2H_2O$) confirmed and expanded these findings. The study demonstrated that 80-85% of sebum palmitate and sapienate (the major human sebum FAs) are derived from DNL, while only 20% of circulating palmitate was DNL-derived, which can only be explained by DNL in SGs playing a major role in sebum synthesis[87]. Notably, meibum synthesis in the human Meibomian gland and sebum synthesis in Syrian hamsters and Göttingen minipigs, widely used model species for studying sebum, seem to be less dependent on DNL[87]. Sebocytes, similarly to blood platelets[88] thus belong to a small group of human cell types that rely on DNL to produce specific lipid pools needed for critical physiological processes.

SG cells are not only extensively equipped with membrane transporters necessary for importing the corresponding substrates, they also operate essentially all enzymatic reactions necessary to synthesize different lipid classes. For instance, mouse SG cells[33] express high levels of the genes encoding DNL enzymes ATP-citrate synthase (*ACLY*)[50] and FA synthase (*FASN*)[77]. In human SGs, both enzymes are highly expressed in SEB-B and more differentiated SEB-1 cells, with lower levels as differentiation progresses; similarly, the gene set "lipid metabolism" has the highest expression in SEB-1 and SEB-2 stages (Fig. 4A). Also, loss of activity of these enzymes may severely affect SG structural integrity and sebum production[58]. DNL in human sebum makes sense considering its unusual FAs containing very long chains, branched moieties, and Δ6 mono-desaturated products[18]. This also suggests that FA oxidation represents an important energy source for ATP production in SGs. While in most cell types FA oxidation concomitant to DNL is largely avoided (futile cycling), sebocytes seem to selectively utilize FAs for β-oxidation or direct incorporation into lipids[89].

## Sebum synthesis relies on peroxisomal activity

Peroxisomes execute numerous metabolic functions (Fig. 3), including catabolism of very long-chain FAs (VLCFA), branched-chain FAs, D-amino acids, and polyamines, and biosynthesis of plasmalogens, peroxisome-derived phospholipids that have a vinyl ether bond at the sn-1 position as opposed to an ester bond in the conventional phospholipids[90]. More recent studies suggested that peroxisomes are critical mediators of cellular responses to various forms of stress, including hypoxia and oxidative stress[91].

Peroxisomes were described decades ago in SG cells in the form of extensive tubular aggregates[92], and it was speculated that the wide variety in peroxisomal complex morphologies may explain the striking differences in sebum composition across species and between single individuals of the same species. Analysis of single-cell RNA data[29] revealed that peroxisomal transcripts are strongly enriched in both mouse and human SGs. Spatial transcriptomics analysis[16] revealed strong expression of peroxisomal genes[90] as sterol carrier protein 2 (*SCP2*) and 3-ketoacyl-CoA thiolase (*ACAA1*) in all sebocyte stages; a similar pattern was observed for the gene set "Peroxisome" (Fig. 4a). Thus, peroxisomes have key but so far rather under-recognized roles in sebaceous lipogenesis.

## Lactate: regulator or waste product?

Besides entry into the TCA cycle as described above, pyruvate can be metabolized to lactate, a process that occurs at high rates in tumors even in the presence of oxygen and a functional oxidative phosphorylation machinery (the *Warburg effect*)[93]. Initially interpreted as a hallmark of cancer cells, aerobic glycolysis is of much broader relevance and may be considered the preferred metabolic program in cases when robust transient responses are needed[94]. Besides being generated by glycolysis, lactate can be derived from glutamine[95]. Cytoplasmic lactate can be exported outside the cell (via SLC16A3) or converted into pyruvate and transported into mitochondria to enter the TCA cycle as acetyl-CoA (Fig. 3). For the latter, cytoplasmic lactate reacts with $NAD^+$ to form pyruvate and NADH. The reaction, catalyzed by the enzyme lactate dehydrogenase (LDH), can take place in both directions: under abundant lactate uptake and adequate oxygenation towards pyruvate; under low oxygen levels, when NADH accumulates and pyruvate cannot be further metabolized, in the lactate direction. Aerobic glycolysis results in less ATP per mole of glucose than does complete glucose oxidation to $CO_2$, raising the question of why metabolically active cells would engage in such a wasteful form of metabolism? First, although the ATP yield per glucose molecule is low, the percentage of cellular ATP it produces can exceed that produced by oxidative phosphorylation, provided that the rate of glycolysis is high. The second advantage of the Warburg effect lies in the fact that many glycolytic intermediates are precursors to anabolic pathways, including NADPH, ribose sugars for nucleotides, or amino acids for proteins[36]. Mechanistically, it was suggested that aerobic glycolysis reflects a metabolic state in which the demand for $NAD^+$ exceeds the demand for ATP[96]. While the reactions that regenerate $NAD^+$ do not directly provide biomass to cells, $NAD^+$ is needed to catabolize reduced nutrients as sugars and lipids, and to synthesize oxidized macromolecules, such as nucleotides and amino acids[97]. Thus, metabolism of pyruvate to lactate also generates an essential co-factor for several metabolic processes. On a more speculative note, aerobic glycolysis in the SG context would allow cells to have fewer mitochondria, thus providing more room for cytoplasmic lipid droplets while at the same time generating $NAD^+$ for multiple metabolic pathways.

Long considered a metabolic waste product, lactate is now recognized as an important exogenous and endogenous carbon source with additional metabolic regulatory functions[98]. Notably, whole-body metabolite analysis in mice revealed that circulating lactate rather than glucose may be a major carbon source fueling the TCA cycle in both fed and fasting mice[99].

Using freshly isolated human SGs, Downie and Kealey showed that about 94% of consumed glucose is converted to lactate[100]. As (peripheral) sebocytes have ample access to oxygen and an intact oxidative phosphorylation machinery, this finding indicates that SGs perform aerobic glycolysis[79]. SLC16A3, which is believed to be the principal lactate exporter[101], has low expression in sebocytes, suggesting that only small amounts of lactate are normally exported. Such a retention of lactate would be compatible with our working hypothesis that SG cells channel all available resources in synthetic processes, but poses the question of how sebocytes handle the potentially enormous amounts of imported or glycolysis-derived lactate. Sebocytes express different LDH isoforms[16,33] and could therefore, theoretically, re-convert lactate to pyruvate. However, this would be a futile cycle, making an alternative mechanism more likely. In fact, it has been shown that lactate can enter mitochondria and be oxidize because of an internal LDH pool[102].

Concerning a regulatory role, histone lactylation is an emerging epigenetic modification that directly stimulates gene transcription from chromatin[103], and lactate was recently identified as a mitochondrial messenger that activates the ETC to increase pyruvate oxidation and lactate utilization[104]. Thus, we conclude that at least part of the imported or glycolysis-derived lactate may have important regulatory functions and/or serve as a substrate for energy production or anabolic processes.

## Glutamine fuels anaplerosis to support sebaceous lipogenesis

Both essential and non-essential amino acids taken up by cells are channeled into a multitude of synthetic pathways. Glutamate, cysteine, and glycine (together with serine as a precursor for the two latter ones) are essential for the synthesis of glutathione (GSH), a multifunctional tripeptide (to be discussed in detail below). Branched chain amino acids leucine, isoleucine, and valine are important protein building blocks, but can also generate TCA cycle intermediates to fuel oxidative ATP generation or a multitude of anabolic processes for the formation of nucleotides or lipids. The list of usages for amino acids seems endless, further including the generation of biogenic amines (as histamine, serotonin, or the catecholamines) and of methyl donors for DNA and RNA methylation[105]. Notably, eighteen of the twenty amino acids in the human body can contribute to gluconeogenesis, with alanine and glutamine being the major contributors[52]. Glutamine, the second most used nutrient in proliferating cells, effectively replenishes intermediates for the TCA cycle (Fig. 3). Although mammals can synthesize glutamine, the cellular demand for it outstrips its supply, and glutamine becomes essential (thus designed as a "conditionally" essential amino acid)[106]. Together with glucose, glutamine satisfies essential cellular needs in the cellular intermediary metabolism, both in the bioenergetic ATP production in the TCA cycle and in the provision of intermediates for macromolecular synthesis. For the former, glutamine exits the TCA cycle as malate, is sequentially converted into pyruvate and acetyl-CoA, before re-entering the cycle. The latter include the synthesis of nucleotides, hexosamines (and thus glycosylation reactions), GSH, and reducing equivalents (essentially NADPH)[106]. The simultaneous usage of both glucose and glutamine is particularly supportive for cells showing the Warburg effect, as glutamine-derived α-ketoglutarate fuels the TCA cycle, an essential component of anaplerotic biomass production. In rapidly proliferating glioblastoma cells, for example, the acetyl-CoA pool is mostly derived from glucose, while the oxalacetate pool is derived essentially from glutamine[107], resulting in citrate molecules (containing carbon derived from both substrates) that can be released as acetyl-CoA to support lipid synthesis.

Sebocytes are equipped with receptors and enzymes to import and process glutamine, and in the absence of glutamine, the rates of proliferation and lipogenesis in humans SGs are reduced by 41% and 37%, respectively[100]. Also, glutamine uptake and metabolism are promoted by CMYC[108], and MYC overactivity in transgenic mice was shown to result in enlarged SGs[58]. Thus, glutamine seems to be an important substrate for sebogenesis.

## Metabolic adaptation to progressive hypoxia

As sebocytes are dislodged towards the gland's middle, they have to cope with two drastic environmental changes: delamination from the basal lamina and exposure to progressively lower levels of oxygen. Hypoxia, which is also a hallmark of solid tumors, typically promotes metabolic rewiring by activating hypoxia-inducible factor 1-alpha (HIF1A), a master transcriptional regulator of the adaptive response to hypoxia. Under hypoxic conditions, HIF1A activates the transcription of over 40 genes, including erythropoietin, glucose transporters, glycolytic enzymes, vascular endothelial growth factor, and other genes whose protein products increase oxygen delivery or facilitate metabolic adaptation to hypoxia. Metastasizing cells are known to form clusters, inducing a hypoxic environment that drives HIF1A-mediated mitophagy, clearing damaged mitochondria and restricting ROS. As a result, hypoxia and reduced mitochondrial capacity promote dependence on glycolysis for ATP production[109]. HIF1A also actively suppresses the TCA cycle by directly transactivating the gene encoding pyruvate dehydrogenase kinase 1 (PDK1) which, among other effects, stimulates aerobic glycolysis and lactate production[110,111]. Another HIF1A target is hypoxia-inducible lipid droplet-associated (HILPDA), a lipid droplet-associated protein that inhibits lipolysis and increases lipid storage stimulating DGAT1-catalyzed TAG synthesis[112]. Notably, a hypoxic microenvironment was shown to increase lipogenesis in cultured human sebocytes[113], and human sebocyte differentiation[16] is accompanied by increased expression of *HILPDA* and UbiA prenyltransferase domain-containing protein 1 (*UBIAD1*), another hypoxia-related gene[114], and the gene set "Hypoxia" also illustrates the sharp increased expression of corresponding genes towards terminal sebocyte differentiation (Fig. 4A). Probably, this hypoxia-induced metabolic switch rewires glucose metabolites from the mitochondria to glycolysis to maintain ATP production and to prevent toxic ROS production (see below).

## Autophagy is a hallmark of terminal sebocyte differentiation

Cells that do not have access to extracellular metabolic substrates, such as cancer cells located in nutrient-deprived microenvironments, frequently engage in the self-catabolic process of macroautophagy[38]. Autophagic cells form autophagosomes, double-membrane vesicles that incorporate organelles and other cytosolic particles and then fuse with lysosomes, leading to the cargo degradation by lysosomal enzymes[115]. Different cancer types upregulate this process, and autophagy inhibition may prevent the growth of solid tumors[38]. Autophagy may sustain cell viability for a limited time period by recycling endogenous macromolecules, but it cannot allow for a sustainable increase in cell mass. Besides this "nutritive function", autophagy sustains cell viability by protecting metabolic active from the deleterious effects of misfolded proteins, toxic lipids, or damaged organelles.

The expression of autophagy-related genes as *WIPI2* and *ATG9B*[116,117] increases as human SG differentiation progresses[16], which can also be observed when assessing the gene set "Autophagy" (Fig. 4A). Autophagy is active in SGs and its suppression by deletion of ATG7 leads to enlarged glands, altered sebum composition, and multiple defects in sebocyte differentiation[118,119]. Autophagy seems to contribute to the maintenance of functional lysosomes in normally differentiating sebocytes, whereas lack of these maintenance and repair mechanisms allows defective lysosomes to trigger cell death via degradation of nuclear DNA by the lysosomal endonuclease DNase2[120,121]. As SG cells are highly metabolic active and later differentiation stages operate in a rather undersupplied environment, autophagy in sebocytes is coherent, and it is reasonable that disrupting autophagy will interfere with the final stages of sebocyte maturation. However, while autophagy is a key component of sebocyte terminal differentiation, it remains to be determined whether it is mechanistically involved in holocrine secretion.

## Progressive ROS-mediated lipid peroxidation

Reduction and oxidation (redox) reactions occur as part of normal cellular metabolism and involve the loss or acquisition of electrons by atoms, generating reactive and mostly short-lived intermediates. The most relevant redox reaction-derived molecules are reactive oxygen species (ROS), comprising free radicals, such as superoxide anion and hydroxyl radical, as well as non-radical species as hydrogen peroxide[122]. The major ROS generators are mitochondria. When electrons are being transferred to oxygen, a part of them leak from the ETC and directly react with oxygen to produce superoxide, the precursor of other ROS[122]. ROS may also be produced in other organelles, such as peroxisomes and the endoplasmic reticulum, and by the activity of numerous enzymes, including NADPH oxidases (NOX), lipoxygenases (LOX), cyclooxygenase (COX), xanthine oxidase (XO), and nitric oxide synthase (NOS)[123].

Maintaining redox homeostasis is essential to all metabolic states, and particularly for cells undergoing proliferation or other intense anabolic processes. At submicromolar concentrations, ROS participate in the control of key cellular functions as cell growth, survival, and differentiation[124]. When ROS levels rise above a critical level, however, they may damage cells by starting chemical chain reactions such as lipid peroxidation, or by oxidizing DNA or proteins. For this reason, spikes of ROS are usually neutralized by a

variety of antioxidant cellular systems, including enzymes as superoxide dismutase (SOD), catalase, glutathione reductase (GR), glutathione peroxidase (GPX), thioredoxins (TRX), and peroxiredoxins (PRX)[122]. Mechanistically, when ROS level rises, kelch-like ECH-associated protein 1 (KEAP1) is oxidized and loses its ability to sequester erythroid 2 like 2 (NRF2), the master regulator of antioxidant response, in the cytosol for proteasomal degradation. The now nuclear NRF2 acts as a transcription factor that activates a collection of genes involved in the synthesis and utilization of cellular antioxidants, including increased expression of the enzyme glutathione peroxidase 4 (GPX4) and transporter proteins and enzymes involved in GSH biosynthesis[125]. GPX4 catalyzes the reduction of lipid peroxides at the expense of reduced GSH. The oxidized form of GSH, which is generated during the reduction of hydroperoxides by GPX4, is recycled by glutathione reductase and $NADPH + H^+$. A tripeptide of glutamate, cysteine and glycine, GSH is a major redox buffer against various sources of oxidative stress[126]. In tumors, GSH is essential for cell survival as it protects cells from oxidative stress associated with intense metabolism, DNA-damaging agents, and inflammation. For instance, glutamine withdrawal caused GSH depletion in fibroblasts with increased MYC activity, and led to frank oxidative damage in hybridoma cells[127]. Maintaining GSH levels is a highly coordinated process that depends on glutamine (to produce glutamate) and glutamate (to acquire cysteine). The latter is mediated by the xCT-Antiporter (a heterodimer formed by SLC3A2 and SLC7A11), which exports glutamate and imports the homodimer cystine, the major source of cysteine for GSH synthesis. SLC7A11, in particular, seems to be upregulated in cancer cells, and its loss reduced the growth of pancreatic tumors[128]. Interestingly, SLC7A11 expression driven by the NRF2 antioxidant program modulates the dependency on other nutrients as glucose and glutamine[129].

One important indicator of cellular redox balance is the maintenance of the NADH to $NAD^+$ ratio. Different substrates, either catabolized in the TCA cycle or diverted into anabolic synthesis, release electrons that are captured by $NAD^+$, leaving the cells with NADH that must be reconverted to $NAD^+$. The continuous passage of electrons through the ETC maintains adenosine triphosphate (ATP) production and also regenerates $NAD^+$. The malate-aspartate and the glycerol-phosphate shuttle promote the translocation of cytosolic electrons into the mitochondria for consumption in oxidative phosphorylation, and are thus important mechanisms to maintain redox homeostasis[125]. In anabolic active cells, the $NADH/NAD^+$ redox pair serves not only to power oxidative phosphorylation, but also supports biosynthesis, as in glycolysis and the TCA cycle, where $NAD^+$ is required to keep the pathway active. Overuse of these pathways with the corresponding oversupply of NADH may overcome the $NAD^+$-regenerating capacity of the ETC, especially when the ultimate electron acceptor, oxygen, becomes limiting.

ROS-induced skin damage has been examined in the context of skin aging[130,131]. In the SG setting, however, researchers have so far focused on the damaging and immunomodulatory effects of environmentally-induced peroxidation of squalene, for instance by ozone or ultraviolet radiation[132,133]. Both phenomena, robust supply of NADH and oxygen scarceness, coincide as sebocytes start synthesizing huge amounts of lipids, with the threat of massive ROS production. In light of their intense lipid metabolism, the most obvious ROS-mediated damage in these cells is lipid peroxidation. Lipid peroxides destabilize membrane structure and fluidity, leading to pore formation and the breakdown of lipid peroxides to toxic aldehydes[134].

SG cells are not only lipid-rich, they also produce a large number of polyunsaturated FAs (PUFAs), whose bis-allylic position methylene groups are particularly vulnerable for hydrogen abstraction and thus lipid peroxide formation. This is likely to explain the protective role of the MUFA oleic acid, as it competes with and thereby reduces the presence of PUFA species in the cellular membranes[135]. Besides PUFAs, SG cells also contain the second essential element for initiating lipid peroxidation, iron, which is required for ATP production in the ETC, besides being a co-factor of enzymes active in sebocytes, including SCD and LDHD. Early stages of sebocyte differentiation, accordingly, exhibit robust transcription of genes

encoding enzymes involved in neutralizing ROS, such as glutathione synthetase (*GSS*)[136] and *SOD1*[137], but their transcription is downregulated towards the terminal stages; this is also suggested by the gene set "ROS metabolic process" (Fig. 4A). Thus, as they mature, sebocytes become particularly vulnerable to ROS and consequently to lipid peroxidation.

## Concluding remarks and outstanding questions
SGs emerge from our analysis as structures with a remarkable ability to take up nutritive substrates from the blood and process them towards secretion products, in particular de novo-generated lipids. Recognizing the *modus operandi* of SG may be instructive for designing new therapeutic approaches for skin diseases as acne, for instance by targeting specific metabolic pathways. Besides the issues already addressed above, there are numerous unanswered questions that will require future studies. Answering these questions (a mixture of obvious and highly speculative ones is summarized below) will be a fascinating and rewarding task.

### Do macropinocytosis and entosis support nutrient acquisition by sebocytes?
Besides the described routes for nutrient acquisition based on transporters, channels, diffusion, or autophagy, SGs may explore additional mechanisms such as macropinocytosis (a non-selective endocytic pathway for bulk ingestion of extracellular solutes) and entosis (engulfment of whole cells)[38,41]. Exploring additional routes would power the SG's ability to take up substrates.

### What's in sebum besides lipids?
While we focused on metabolic pathways leading to lipid synthesis, sebum includes additional components such as antimicrobial peptides[8] and volatile substances[138]. Systematic assessment of sebum composition (e.g., metabolomics) has been surprisingly underexplored and is likely to shed light on so far unrecognized messengers in sebum.

### Do SG cross-talk with key metabolic organs?
SGs are under the influence of several hormones, growth factors, and cytokines[7,8]. While the ability of sebocytes to synthesize a myriad of hormones has been postulated[139], a functional relevance for them has never been documented.

In a broader context, the role of lipid metabolism along various axes, such as the liver-brain and liver-gut axes, as well as its connection to the skin in the context of the immune system, requires further investigation[140]. The association between skin diseases, such as acne, and pathogenic lipid metabolism in the SGs, along with links to TREM2-mediated macrophage reprogramming, is well established[141]. APOE, a protein highly expressed in the liver and central to LDL transport through the bloodstream, also shows significant gene expression in the basal layer of SGs and in melanomas, where it induces phenotype transitions of melanoma cells between states of varying invasiveness[142]. Moreover, APOE has been identified as a marker for metastatic colorectal cancer, where it facilitates intercellular communication between macrophages and fibroblasts, thereby promoting colorectal cancer progression in the immunosuppressed metastatic niche within the liver[143]. Collectively, these findings suggest a systemic crosstalk of lipid metabolic pathways across the body including SG, liver, gut, and possibly other organs as the adipose tissue. Combining endocrinological methods and transcriptomics/proteomics data is a promising approach to study whether and how SGs communicate with organs as muscle, liver, and adipose tissue.

### How does the sebocyte handle metabolism-derived intermediates?
Hyperactive cellular metabolism frequently results in incomplete breakdown and accumulation of intermediates. If sebocytes are programmed to make use of all available substrates, excretion of such intermediates is an unattractive option. A quantitatively important intermediate is pyruvate. As discussed above, if its usage in the TCA cycle is saturated, it may be converted to lactate, which may have regulatory functions. Another possible fate

of pyruvate is acetate, which may be mediated by ROS activity or via keto acid dehydrogenase activity[50]. The generated acetate can then be converted to acetyl-CoA and contribute to lipogenesis (indicated in Fig. 3). Notably, this indicates that pyruvate may function as a natural ROS scavenger. However, this usage of lactate and acetate by sebocytes remains to be confirmed, and the same applies to the handling of other products of cellular metabolism (e.g., formic acid).

## What is the mechanistic basis of holocrine secretion?

While there is robust evidence that autophagy and lysosomal degradation of cellular components participate in the cell death during terminal differentiation of sebocytes, the exact sequence of events leading to holocrine secretion has not been characterized. As sebaceous lipogenesis proceeds, the intense metabolic activity imposes a cellular burden (ROS-mediated lipid peroxidation, ER stress) that is likely to compromise viability. To put it simple, sebocytes work themselves to death. Thus, holocrine secretion seems to combines both "active" processes (autophagy, lysosomal degradation) and a "passive" component, the progressive non-maintenance of vital cellular functions, in particular the protection from oxidative stress.

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

## Acknowledgements
M.S. and H.B. express gratitude to the German Ministry for Science and Education (BMBF) and State Committee of Science of Armenia (SCS) for the financial support under grant numbers 01DK24004 (BMBF) and 22SC-BMBF-1C004 and 21AG-1F021 (SCS).

## Author contributions
M.S.: figure drafting, writing—original draft. H.B.: figure drafting, writing—original draft. M.R.S.: conceptualization, supervision, figure drafting, writing—original draft.

## Funding

## Competing interests
The authors declare no competing interests.
