## [Transparent Peer Review file · Communications Biology]

The metabolic underpinnings of sebaceous lipogenesis

Corresponding Author: Professor Marlon Schneider

Version 0:

Reviewer comments:

Reviewer #1

(Remarks to the Author)

The authors provide a very detailed, comprehensive and up to date review of sebaceous gland metabolism.

1. The structure of the review in some places could confuse the reader especially with regards aerobic glycolysis and lactate production which I would consider a defining feature of sebaceous gland metabolism. The authors bring this into their review but much later under 'Lactate regulator or waste product' I think at least some mention of aerobic glycolysis needs to come earlier even if the authors then reference that this is covered in more detail later.

Reason I suggest this is the authors describe classical metabolic pathways with Glucose being metabolized to pyruvate and then acetyl CoA entering TCA cycle. But as Downie et al have shown only 6% of glucose is oxidized and most is metabolized to Lactate. I appreciate its a challenge as the section on Lactate is nicely written

2. In their discussion of 'Lactate regulator or waste product' the authors mention the conundrum surrounding why a tissue would engage in aerobic glycolysis which initially appears wasteful. They mention off course that metabolism of pyruvate to lactate generates NAD+. It may be nice to expand on this slightly and mention that off course NAD+ is essential for biosynthesis. They may also like reference work suggesting that in some proliferating tissues aerobic glycolysis reflects a metabolic state in which the demand for NAD+ exceeds the demand for ATP (Luengo et al 2021 Molecular Cell 81 691-707)

3. In some tissues where glucose is not limiting glucose may be preferentially metabolized to lactate as the cell may in fact be limited by the number of mitochondria it can accommodate. I wonder whether as the sebocyte basically fills itself with lipid its preferential to have fewer mitochondria and generate ATP via aerobic glycolysis which also generates NAD+ for biosynthesis. See Vaupel 2021 (J Physiol. 2021;599(6):1745–57)

Reviewer #2

(Remarks to the Author)

This is a review paper that discusses the mechanistic detail of energy metabolism and lipid production, in general physiology, but also specifically within the sebocyte/sebaceous gland.

In general the manuscript is exceptionally well-written, complete and offers a unique prospective on lipogenesis/lipid metabolism in sebaceous glands.

Portions of the manuscript are exceptionally broad--for example, there is an initial discussion on obesity, but then no discussion on how obesity is related to lipid metabolism or sebaceous glands, making the discussion out of place for the focus of the manuscript (mechanisms of lipogenesis in sebocytes). I would suggest removing the obesity discussion or making a stronger connection to the thesis of the manuscript.

The metabolic underpinnings of sebaceous lipogenesis

Manuscript COMMSBIO-25-1757

Point by point response to the comments of the reviewers

Reviewer 1

The authors provide a very detailed, comprehensive and up to date review of sebaceous gland metabolism.

1. The structure of the review in some places could confuse the reader especially with regards aerobic glycolysis and lactate production which I would consider a defining feature of sebaceous gland metabolism. The authors bring this into their review but much later under 'Lactate regulator or waste product' I think at least some mention of aerobic glycolysis needs to come earlier even if the authors then reference that this is covered in more detail later.

Reason I suggest this is the authors describe classical metabolic pathways with Glucose being metabolized to pyruvate and then acetyl CoA entering TCA cycle. But as Downie et al have shown only 6% of glucose is oxidized and most is metabolized to Lactate. I appreciate its a challenge as the section on Lactate is nicely written

Answer: We thank the reviewer for this suggestion. We now introduce the concept of “aerobic glycolysis” much earlier in the manuscript (page 7) and mention that this aspect will be discussed in details at the appropriate section (pages 14-15).

2. In their discussion of 'Lactate regulator or waste product' the authors mention the conundrum surrounding why a tissue would engage in aerobic glycolysis which initially appears wasteful. They mention off course that metabolism of pyruvate to lactate generates NAD+. It may be nice to expand on this slightly and mention that off course NAD+ is essential for biosynthesis. They may also like reference work suggesting that in some proliferating tissues aerobic glycolysis reflects a metabolic state in which the demand for NAD+ exceeds the demand for ATP (Luengo et al 2021 Molecular Cell 81 691-707)

Answer: This is a very good point and we now clearly mention that NAD+ is essential for several biosynthetic pathways, provide examples and cite an appropriate reference. We also include the mechanistic explanation suggested by Luengo et al. Both amendments can be found on page 14.

3. In some tissues where glucose is not limiting glucose may be preferentially metabolized to lactate as the cell may in fact be limited by the number of mitochondria it can accommodate. I wonder whether as the sebocyte basically fills itself with lipid its preferential to have fewer mitochondria and generate ATP via aerobic glycolysis which also generates NAD+ for biosynthesis. See Vaupel 2021 (J Physiol. 2021;599(6):1745–57)

Answer: We thank the reviewer for raising this intriguing point. First, we now cite the publication by Vaupel & Multhoff when mentioning the Warburg effect for the first time (page 14). Second, we make clear that the Warburg effect is not restricted to cancer cells, but the “preferred metabolic program when robust transient responses are needed”, and cite an appropriate reference (page 14). Third, we now mention the intriguing idea that sebocytes may restrict mitochondria to allow accumulation of a larger number of lipid droplets in the cytoplasm, which at the same times generates NAD+ for biosynthesis (page 15).

Reviewer 2

This is a review paper that discusses the mechanistic detail of energy metabolism and lipid production, in general physiology, but also specifically within the sebocyte/sebaceous gland.

In general the manuscript is exceptionally well-written, complete and offers a unique prospective on lipogenesis/lipid metabolism in sebaceous glands.

Portions of the manuscript are exceptionally broad--for example, there is an initial discussion on obesity, but then no discussion on how obesity is related to lipid metabolism or sebaceous glands, making the discussion out of place for the focus of the manuscript (mechanisms of lipogenesis in sebocytes). I would suggest removing the obesity discussion or making a stronger connection to the thesis of the manuscript. Answer: We thank the reviewer for assessing our article and for his suggestions. We understand the point of the reviewer and have shortened the obesity discussion. We feel however, that comparison of the sebaceous gland with the adipose tissue provides a nice introduction to the topic. Thus, we maintained the comparison in the text and in Fig. 1B.